# Modeling Growth Kinetics of *Escherichia coli* and Background Microflora in Hydroponically Grown Lettuce

**DOI:** 10.3390/foods13091359

**Published:** 2024-04-28

**Authors:** Xiaoyan You, Dongqun Yang, Yang Qu, Mingming Guo, Yangping Zhang, Xiaoyan Zhao, Yujuan Suo

**Affiliations:** 1Henan Engineering Research Center of Food Microbiology, College of Food and Bioengineering, Henan University of Science and Technology, Luoyang 471023, China; xiaoyanyou@haust.edu.cn (X.Y.); ydq1680175@163.com (D.Y.); 2Laboratory of Quality and Safety Risk Assessment for Agro-Products of Ministry of Agriculture and Rural Affairs, Institute for Agro-Food Standards and Testing Technology, Shanghai Academy of Agricultural Sciences, Shanghai 201403, China; quyang@saas.sh.cn (Y.Q.);; 3Zhejiang Key Laboratory for Agricultural Food Process, College of Biosystems Engineering and Food Science, Zhejiang University, Hangzhou 310058, China; 4Shanghai Leafa Agriculture Development Co., Ltd., Shanghai 201203, China; hortizhang@163.com

**Keywords:** growth model, hydroponically grown lettuce, *Escherichia coli*, background microflora

## Abstract

Hydroponic cultivation of lettuce is an increasingly popular sustainable agricultural technique. However, *Escherichia coli*, a prevalent bacterium, poses significant concerns for the quality and safety of hydroponically grown lettuce. This study aimed to develop a growth model for *E. coli* and background microflora in hydroponically grown lettuce. The experiment involved inoculating hydroponically grown lettuce with *E. coli* and incubated at 4, 10, 15, 25, 30, 36 °C. Growth models for *E. coli* and background microflora were then developed using Origin 2022 (9.9) and IPMP 2013 software and validated at 5 °C and 20 °C by calculating root mean square errors (RMSEs). The result showed that *E. coli* was unable to grow at 4 °C and the SGompertz model was determined as the most appropriate primary model. From this primary model, the Ratkowsky square root model and polynomial model were derived as secondary models for *E. coli*-R168 and background microflora, respectively. These secondary models determined that the minimum temperature (T_min_) required for the growth of *E. coli* and background microflora in hydroponically grown lettuce was 6.1 °C and 8.7 °C, respectively. Moreover, the RMSE values ranged from 0.11 to 0.24 CFU/g, indicating that the models and their associated kinetic parameters accurately represented the proliferation of *E. coli* and background microflora in hydroponically grown lettuce.

## 1. Introduction

Hydroponic agriculture, notable for its enhanced yield per acre and diminished consumption of resources, is gaining recognition as a sustainable alternative to conventional soil-based farming, offering a solution to pressing challenges of global food security [1]. In 2022, the Food and Agriculture Organization of the United Nations (FAO) published a report supporting the cultivation of vegetables through hydroponics methods, predicting that hydroponically grown lettuce would become one of the leading green leafy vegetables in terms of sales [2]. However, while hydroponic greenhouses facilitate year-round production by controlling environmental conditions such as temperature and humidity, they inadvertently create an environment that is conducive to the persistence and transmission of foodborne pathogens, thereby posing potential health risks.

The incidence of infections resulting from the contamination of hydroponically grown produce with foodborne pathogens has been documented. A deadly outbreak of *Escherichia coli* O104 occurred in Germany in 2011, where hydroponically grown fenugreek sprouts served as the vehicle of transmission, resulting in more than 4000 infections and 53 deaths [3]. In 2014, the United States encountered outbreaks of illness attributable to the contamination of hydroponic sprouts with *E. coli* O157, followed by numerous outbreaks of *Salmonella* infections associated with hydroponically cultivated lettuce in 2021 [4]. These incidents highlighted the persistent risks associated with hydroponic leafy greens. Moreover, since 2006, it has been estimated that around 7000 individuals in the United States fall ill annually due to the consumption of lettuce contaminated with *E. coli*, a Gram-negative bacterium that significantly compromises the safety and quality of this produce [5,6,7].

*E. coli* is recognized for specific serotypes that can pose serious health risks to humans. Owing to the challenges in detecting pathogenic serotypes, the enumeration method for *E. coli* is widely employed in food products as an indicator of sanitary quality. This approach focuses on the assessment of product safety based on concentration levels of *E. coli*, rather than identifying the pathogenicity of specific *E. coli* strains. Furthermore, *E. coli* serves as a crucial hygienic indicator in the risk assessment of foodborne pathogens. The risk associated with a particular pathogen can be gauged by comparing its contamination ratio to that of *E. coli* [8,9,10].

Models that predict the proliferation of microorganisms in food play a pivotal role in evaluating and managing both food quantity and public health risks. Although numerous studies have focused on formulating growth models for *E. coli* in fresh-cut lettuce, the applicability of these models to lettuce cultivated hydroponically may be limited [11,12]. Firstly, the existing models were developed using sterilized samples of fresh-cut leaf, neglecting the influence of background microflora and the integrity of the leaf on *E. coli* growth. Secondly, these models are predicated on lettuce grown in soil, disregarding the fact that the nutritional quality characteristics of lettuce grown hydroponically could influence the growth of microorganisms, including *E. coli* [13]. In the processing of lettuce, maintaining the hygiene of natural lettuce is crucial for ensuring the quality and safety of the product [11]. Therefore, the development of predictive models for the growth of *E. coli* in hydroponically grown lettuce under natural conditions is valuable for predicting shelf life and conducting quantitative risk assessments of *E. coli*. Nevertheless, research focusing on uncut leaves of lettuce remains scarce.

In this study, the growth kinetics of *E. coli* was investigated using leafy greens in their natural state. The objective of this study was to develop models capable of delineating the growth patterns of *E. coli* in lettuce cultivated hydroponically under different temperature conditions. The models derived from this research are anticipated to offer valuable insights for mitigating the proliferation of *E. coli* in hydroponically grown lettuce and facilitating risk assessment processes.

## 2. Materials and Methods

### 2.1. Bacterial Cultures and Preparation

A strain of *E. coli*, designated as *E. coli*-R168, possesses resistance to rifampicin (rif, Beijing Solarbio Science & Technology Co., Ltd., Beijing, China) and was used in this study. The original strain was isolated from lettuce and was artificially induced in our laboratory to develop resistance against 100 mg/L rifampicin. Bacterial cultures with 15% glycerin were stored at −80 °C. The frozen cultures were activated and propagated in tryptone soya broth (TSB, Termo Fisher Scientific, Waltham, MA, USA) plates supplemented with 100 mg/L rifampicin (TSB/R) by aerobic incubation at 37 °C for about 12 h. A single representative colony was inoculated into 10 mL TSB containing 100 mg/L rifampicin. The strain was incubated at 37 °C (190 rpm) for approximately 12–13 h to achieve the stationary phase. Then, it was centrifuged (8000 rpm, 10 min at 4 °C), the supernatant was discarded, and sterile buffer peptone water (BPW, Termo Fisher Scientific, Waltham, MA, USA) was added.

### 2.2. Sample Preparation and Inoculation

The hydroponically grown lettuce samples were obtained from Shanghai Green Cube Agricultural Development Co., Ltd., Shanghai, China. Fresh and intact leaves were chosen, with each portion weighing 10 g, and then placed in sterile homogeneous bags (BKMAM^®^, 200 mm × 270 mm, Changde BKMAM Biotechnology Co., Ltd., Changde, China). The non-homogenized samples were transferred to a biological safety cabinet, where 500 µL aliquots of the cultures were spread on the surface of each lettuce sample uniformly using a micropipette and a spreader, resulting in an initial inoculum level of around 10–100 CFU/g [14]. Following inoculation, the samples are dried aseptically for 15 min before sealing the homogenized bags.

### 2.3. Growth Studies

The inoculated samples were incubated in incubators maintained at constant temperatures (4, 10, 15, 20, 25, 30, and 36 °C). To determine the bacterial populations for constructing growth curves, samples were withdrawn from incubators at different time intervals according to the varying temperatures. To each sample, 90 mL of BPW was added and subjected to pulsation for 2 min at the highest speed in a stomacher (Model BagMixer^®^ 400 W, Interscience Co., Saint Nom, France). An aliquot (0.1 mL) of the liquid portion was withdrawn and plated onto TSB or TSB/R agar plates after proper serial dilution with BPW. After incubation at 37 °C for 24 or 48 ± 2 h, the colonies of *E. coli* and background microflora on each agar were counted. The difference in colony counts between the TSB and TSB/R plates was considered as the count of background microflora. All experiments were repeated independently twice, with each dilution performed in triplicate.

### 2.4. Primary Model Development

The growth kinetic parameters of microorganisms in lettuce were calculated using four primary models based on the temperature-dependent growth of *E. coli* and background microflora. The models employed were: SGompertz, Huang, Berenyi, and SLogistic [15,16,17]. The SGompertz and SLogistic models were from Origin 2022 software, while the Huang and Berenyi models were obtained from IPMP 2013 software [18]. These models were utilized to describe the growth curves, yielding the maximum growth rate (*μ*_max_, Log CFU/g/h) and lag phase duration (*λ*, h).

### 2.5. Secondary Model Development

The secondary models show the effect of temperature changes on the parameters (*μ*_max_, *λ*) of *E. coli* and background microflora in samples, respectively. These models were built using the Ratkowsky square root model (Equation (1)), polynomial model (Equation (2)), and the inverse second order polynomial model (Equation (3)) [19,20,21].
(1)$$\sqrt{\mu_{\max}}=a*\left(T-T_{\min}\right)$$
(2)$$\mu_{\max}=a+bT+cT^{2}$$
(3)$$\mu_{\max}=a+b\left(1/T\right)+c{\left(1/T\right)}^{2}$$
where *T* is the storage temperature (°C), *T*_min_ is the theoretical lowest bacterial growth temperature (°C), *µ*_max_ is the maximum specific bacterial growth rate (h^−1^), *a*, *b*, *c* are the regression coefficients.

For the lag time, a linear regression equation was used to analyze *λ* as a function of *µ*_max_; *a* and *b* are regression coefficients [22].
(4)$$Ln(\lambda )=a-b*Ln(\mu_{\max})$$

### 2.6. Validation of Predictive Models

The validity of the growth prediction model was confirmed by calculating the root mean square error (RMSE). This validation process involved comparing the parameters derived from the secondary model to the parameters through the conventional growth measurement method (not using temperature for model development) (5 °C and 20 °C). The *E. coli* growth data we obtained from ComBase are specific to fresh-cut lettuce leaves at temperatures of 5 °C and 20 °C.

## 3. Results and Discussion

### 3.1. The Primary Model of E. coli-R168 and Background Microflora

#### 3.1.1. The Primary Model of *E. coli*-R168

Hydroponically grown lettuce was inoculated with *E. coli*-R168 and incubated at temperatures ranging from 4 to 36 °C. While studies have indicated that *E. coli* does not proliferate in soya bean products and ground chicken meat at 4 °C, there is potential variability in the minimum growth temperature of microorganisms in different food substrates [23,24]. Considering that 4 °C is a common refrigeration temperature, we initiated our investigation on *E. coli* growth from this temperature. The results revealed that *E. coli* also did not exhibit growth in hydroponic lettuce at 4 °C (Figure 1). Subsequently, the temperature was increased, and predictive models were developed to describe the growth of *E. coli*-R168 at 10, 15, 25, 30, and 36 °C. The results illustrated in Figure 2A exhibit a concurrence between the observed values and the predictions of the SGompertz, Huang, Baranyi, and SLogistic models, indicating that these models successfully describe the growth pattern of *E. coli*-R168 growth on leaf of lettuce. To compare the accuracy of the models, three key parameters were analyzed for each model: Akaike information criterion (AIC), RMSE, and coefficient of determination (R^2^). Although the R^2^ values for all four models exceeded 0.9, the SGompertz model exhibited the smallest AIC and RMSE parameters, which ranged from −23.206 to −0.60484 and 0.053 to 0.109, respectively. Lower AIC values indicate superior model quality [25], while a lower RMSE value closer to 0 signifies better alignment between model predictions and experimental data [26]. The R^2^ value, ranging from 0 to 1, serves as an overall measure of prediction accuracy, with a value of 1 indicating optimal model performance [27] (Table 1). Consequently, the SGompertz model was identified as the most suitable growth model for *E. coli*-R168 in this experiment.

The study revealed that the maximum growth rates (*µ*_max_) of *E. coli*-R168 in the SGompertz model at 10, 15, 20, 25, 30, and 36 °C were 0.020, 0.060, 0.382, 0.586, and 0.875 Log CFU/g/h, respectively (Table 1). As the temperature increased, the growth rate of *E. coli*-R168 also increased, while the lag period decreased, aligning with previous research below 42 °C from Katipoglu-Yazan, T etc. [28] (Figure 2A and Table 1). When comparing the specific growth rates reported by Kim, Y. J [29] and de Oliveira Elias, S. [30] for fresh-cut lettuce at 5–42 °C (Figure 3), the growth rate of *E. coli*-R168 in this study was found to be higher, which might be attributed to the influence of background microflora and the type of lettuce used [31].

#### 3.1.2. The Primary Model of Background Microflora

The background microflora exhibited an irregular and slow growth trend at 4 °C (Figure 1). It has been shown that the bacterial communities residing on the leaves of hydroponically grown lettuce are intricately complex. Low temperatures have significant impacts on the biological structure, which in turn affects the growth of these bacterial colonies [32]. Hence, data collected at 10, 15, 25, 30, and 36 °C were utilized to develop the primary growth model of background microflora in hydroponically grown lettuce (Figure 2B). The Huang, Baranyi, and SGompertz models exhibited superior fit to the growth data of background microflora in comparison to the SLogistic model. A comprehensive evaluation using the parameters of AIC, RMSE, and R2 as presented in Table 1 revealed that the SGompertz model outperformed the others with the lowest AIC values (−12.278 to −42.200) and RMSE values (0.111 to 0.216 CFU/g), coupled with an acceptable R2 value. On this basis, the SGompertz model was selected as the primary model for describing the growth kinetics of background microflora in hydroponically grown lettuce. The maximum growth rates (*µ*_max_) of background microflora, as described by the SGompertz model at temperatures of 10, 15, 20, 25, 30, and 36 °C, were 0.020, 0.086, 0.282, 0.274, and 0.432 h^−1^ (Table 1), respectively.

In this study, the growth rates of background microflora in hydroponically grown lettuce increased at a slower rate compared to *E. coli* when temperatures rose (Figure 3). This phenomenon could be due to the complex composition of local microbes and their interactions within the hydroponically grown lettuce environment [33]. Moreover, other research has shown that in raw ground pork, the growth rate of the background microflora is higher than that of *E. coli*, highlighting the impact of varying nutritional compositions of food matrices on microbial growth [32]. Overall, the results of this study revealed that the growth of *E. coli* in hydroponically grown lettuce was not significantly inhibited by the presence of background microflora.

**Figure 2 foods-13-01359-f002:**
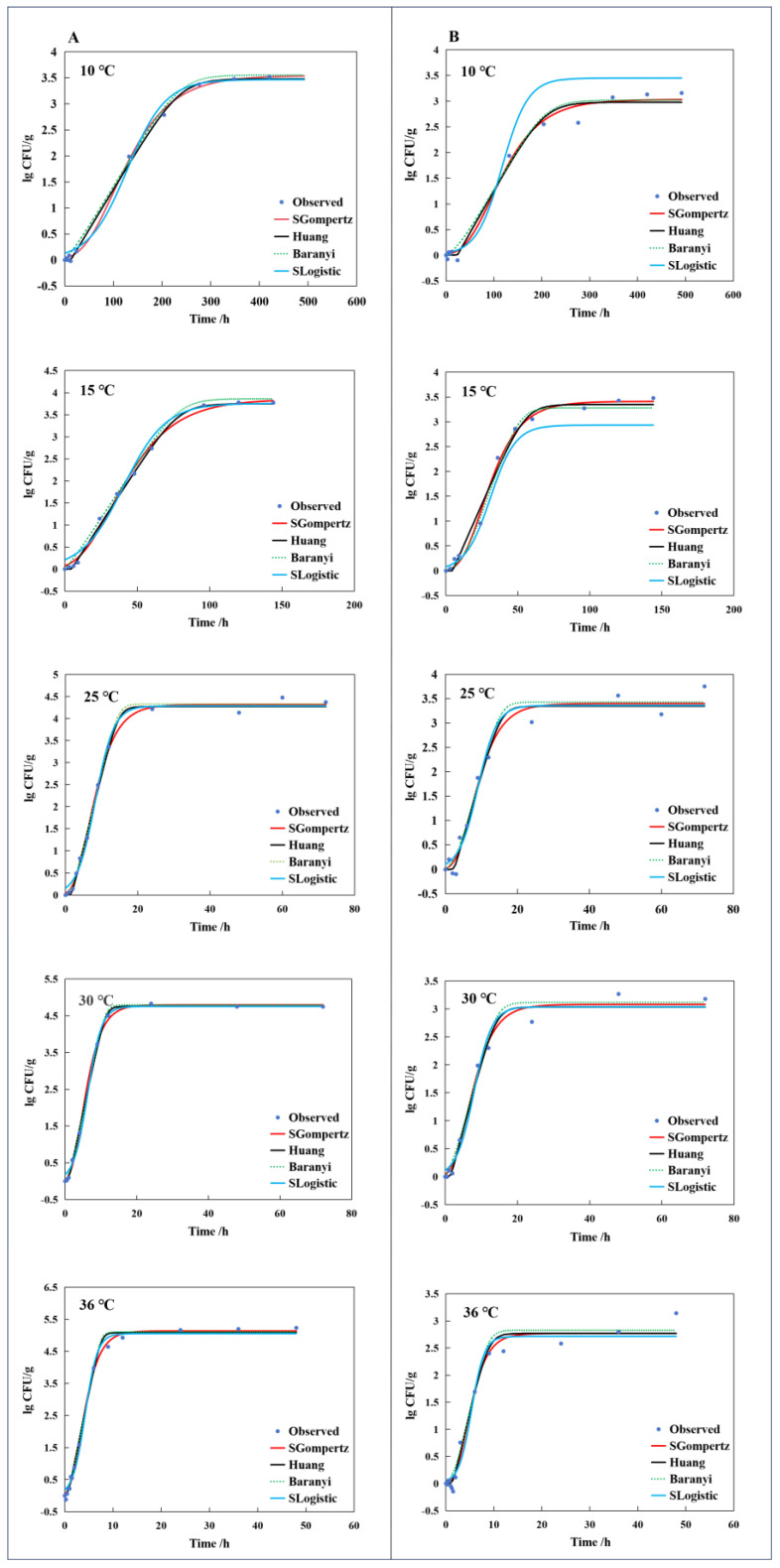
Growth curves of *E. coil*-R168 (**A**) and background microflora (**B**) in hydroponically grown lettuce at 10−36 °C.

**Figure 3 foods-13-01359-f003:**
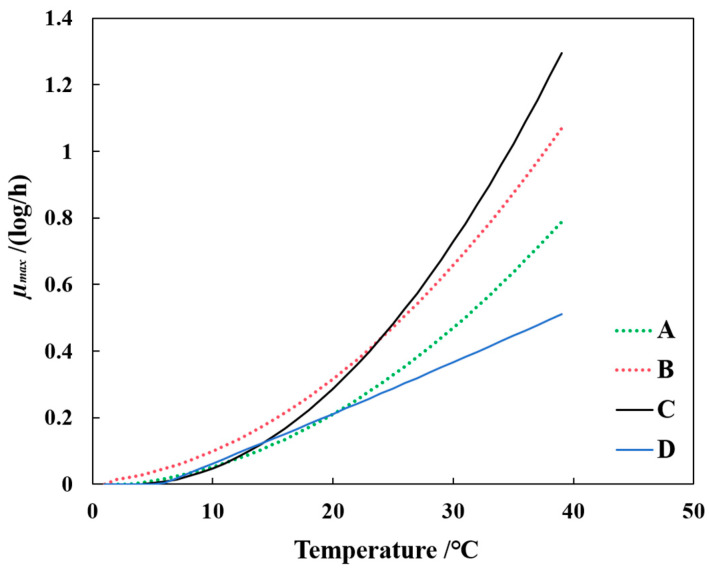
Effect of temperature on specific growth rate and lag times of *E. coil*-R168 and background microflora in hydroponically lettuce and comparison with data from fresh-cut lettuce. Data sources: A = Kim, Y. J. [27]; B = de Oliveira Elias, S. [28]; *E. coil*-R168 (C) and background microflora (D) used in this study.

**Table 1 foods-13-01359-t001:** Fitting parameters of four primary kinetic models of *E. coli*-R168 and background microflora in hydroponically grown lettuce.

Temperature/°C	Model	*μ*_max_/h^−1^	*λ*/h	AIC	RMSE	R^2^
*E. coli*-R168						
10	SGompertz	0.020	35.682	−56.906	0.053	0.999
	Huang	0.036	12.267	−30.439	0.150	0.999
	Baranyi	0.036	9.667	−25.072	0.187	0.998
	SLogistic	0.022	49.551	−35.939	0.126	0.991
15	SGompertz	0.060	8.889	−60.484	0.109	0.993
	Huang	0.117	4.601	−20.644	0.180	0.998
	Baranyi	0.117	2.385	−15.890	0.224	0.997
	SLogistic	0.065	13.274	−27.890	0.479	0.983
25	SGompertz	0.382	2.337	−42.701	0.095	0.997
	Huang	0.758	1.706	−17.101	0.261	0.997
	Baranyi	0.798	1.930	−16.681	0.266	0.997
	SLogistic	0.423	3.229	−34.905	0.132	0.993
30	SGompertz	0.586	1.682	−23.206	0.077	0.998
	Huang	1.047	0.891	−15.886	0.167	0.999
	Baranyi	1.088	1.070	−15.410	0.171	0.999
	SLogistic	0.612	2.395	−12.564	0.132	0.995
36	SGompertz	0.875	1.112	−56.148	0.103	0.997
	Huang	1.688	0.620	−17.423	0.383	0.996
	Baranyi	1.763	0.768	−18.130	0.374	0.996
	SLogistic	0.989	1.669	−40.384	0.174	0.993
Background microflora						
10	SGompertz	0.020	38.403	−34.609	0.181	0.993
	Huang	0.038	23.825	−5.286	0.427	0.987
	Baranyi	0.038	20.763	−3.102	0.468	0.997
	SLogistic	0.027	63.579	−9.191	0.531	0.982
15	SGompertz	0.086	10.400	−33.632	0.111	0.992
	Huang	0.152	4.672	−2.169	0.418	0.988
	Baranyi	0.191	11.267	−5.597	0.358	0.991
	SLogistic	0.096	13.529	−12.122	0.296	0.994
25	SGompertz	0.282	2.864	−23.037	0.216	0.973
	Huang	0.595	2.512	3.213	0.609	0.976
	Baranyi	0.637	2.733	4.975	0.655	0.972
	SLogistic	0.321	3.824	−19.458	0.251	0.963
30	SGompertz	0.274	1.985	−12.278	0.133	0.987
	Huang	0.560	1.534	2.982	0.429	0.987
	Baranyi	0.575	1.473	4.043	0.453	0.985
	SLogistic	0.038	3.069	−6.472	0.178	0.977
36	SGompertz	0.432	1.937	−42.200	0.164	0.980
	Huang	0.864	1.449	−14.170	0.427	0.982
	Baranyi	0.956	1.765	−10.791	0.478	0.978
	SLogistic	0.514	2.708	−36.513	0.198	0.970

### 3.2. Secondary Model of E. coli-R168 and Background Microflora

To investigate the impact of temperature on the growth parameters of *E. coli*-R168 and background microflora, three models were analyzed to determine the optimal secondary model based on the accuracy factor (*A_f_*), bias factor (*B_f_*), RMSE, and R^2^ values [34]. The results from Table 2 indicate that the Ratkowsky square root model offers a superior fit for the growth rate of *E. coli*, with parameter values of *A_f_* = 1.144, *B_f_* = 0.989, RMSE = 0.018, and R^2^ = 0.995. In comparison, other models did not perform as well. Additionally, the polynomial model effectively described the relationship between the growth rate and temperature for background microflora, with *A_f_* = 1.091, *B_f_* = 1.000, RMSE = 0.028, and R^2^ = 0.963. According to the established criteria, *A_f_* and *B_f_* values within the range of 0.9–1.05 are considered ideal, while values from 1.06–1.15 are also deemed acceptable. Values outside this range are considered unacceptable [35]. Therefore, the Ratkowsky square root model and polynomial model were chosen as the secondary models for *E. coli*-R168 and background microflora, respectively. The secondary model revealed that the minimum growth temperature of *E. coli*-R168 in hydroponically grown lettuce was 6.1 °C, which differed by 1–2 °C compared to other foods like raw beef and ground chicken in various media [24,36]. The growth of *E. coli*-R168 was inhibited at a storage temperature lower than 6 °C, and the retardation period was as long as 600 h. At a temperature of 25 °C, after 3.3 h the *E. coli*-R168 growth increases, posing a higher risk of foodborne illness. A linear relationship was observed between Ln *μ*_max_ of *E. coli*-R168 in hydroponically grown lettuce and Ln *λ*, where higher growth rates corresponded to shorter lag periods. Therefore, controlling the lag period through temperature regulation is crucial.

Compared to *E. coli*-R168, the minimum growth temperature of the background microflora in hydroponically grown lettuce was 8.7 °C (Figure 4B), which was higher than that of *E. coli*-R168. The linear relationship between the growth rate and the lag period of background microflora was also similar to that of *E. coli* (Figure 4B). The lag phase duration of the background microflora decreased gradually with an increase in the growth rate. This study, combined with the minimum growth temperature of *E. coli*-R168, posits that maintaining hydroponically grown lettuce storage temperatures below 6 °C can effectively inhibit the growth of *E. coli* as well as native microbial populations, thereby reducing the risk of foodborne diseases.

### 3.3. Model Validation

The secondary model revealed that the minimum growth temperature for *E. coli*-R168 is 6.101 °C. To validate the growth of *E. coli*-R168 in hydroponically grown lettuce at low temperatures, data from the ComBase database were used for external verification to confirm the growth of *E. coli*-R168 at 5 °C. The model’s predictions were consistent with the database’s recording, showing no significant growth of *E. coli*-R168 at 5 °C (Figure 5). Studies have indicated that transportation temperatures exceeding 5 °C can increase the risk of spoilage in agricultural products, potentially leading to disease [37,38]. Therefore, it is crucial to maintain temperatures below 5 °C. In addition, considering the temperature of lettuce at farmers’ markets, an internal verification temperature of 20 °C was chosen, which was not included in the modeling process. The comparison between predicted and observed growth curves is illustrated in Figure 5. The disparity between observed values and the model’s predictions is quantified by the RMSE, with values closer to 0 indicating a more precise alignment between the model’s predictions and actual results. The performance of the *E. coli*-R168 models is documented in Table 3, with RMSE values of 0.1145 at 5 °C and 0.239 at 20 °C. RMSE values ranging from 0.2 and 0.5 are generally considered as normal experimental error [39,40], suggesting that the predictive model constructed in this study is accurate.

Due to variations in the background microflora composition among species, there are no corresponding microbial growth data available in the ComBase database. Therefore, the growth model for background microflora was solely validated at 20 °C (Figure 5). The RMSE for background microflora at 20 °C, as listed in Table 3, was 0.145, indicating that the developed model effectively describes the growth of background microflora in hydroponically grown lettuce.

## 4. Conclusions

This study has successfully developed and validated predictive growth models for *E. coli*-R168 and background microflora in hydroponically grown lettuce, offering significant insights into the microbial safety of such produce. The SGompertz model emerged as the most suitable primary model for describing the growth of *E. coli*-R168, with the Ratkowsky square root model and polynomial model serving effectively as secondary models for *E. coli*-R168 and background microflora, respectively. The primary prediction model indicated that, within the temperature range of 10–36 °C, the hysteresis periods for *E. coli* and the background microflora varied from 35.7 to 1.1 h and 38.4 to 1.9 h, respectively. The secondary models have elucidated the minimum growth temperatures for *E. coli*-R168 and background microflora to be 6.1 °C and 8.7 °C, respectively. These models provide a valuable tool for predicting the shelf life of hydroponically grown lettuce and contributing to quantitative microbial risk assessment.

However, it is important to acknowledge the study’s limitations. The focus on a single strain of *E. coli* and the exclusion of other pathogens that might affect hydroponically grown lettuce could limit the broader applicability of our findings. Future research should aim to include a wider range of pathogens to provide a more comprehensive understanding of microbial risks in hydroponic agriculture. Additionally, while the models were validated at specific temperatures, extending this validation across a broader range of real-world storage and transportation conditions could enhance the practical utility of our findings. The study, which selected an *E. coli* isolate from lettuce for modeling in order to minimize the impact of strain heterogeneity on model accuracy, would benefit from further validation using different types of *E. coli* to enhance the model’s precision.

In conclusion, while our study provides important insights into the growth patterns of *E. coli* and background microflora in hydroponically grown lettuce, it also highlights the need for further research to broaden the scope, validation, and applicability of our findings. By addressing these limitations, future studies can contribute to the development of more comprehensive strategies for ensuring the microbial safety of hydroponically grown produce, thereby supporting the sustainability and public health goals of modern agriculture.

## Figures and Tables

**Figure 1 foods-13-01359-f001:**
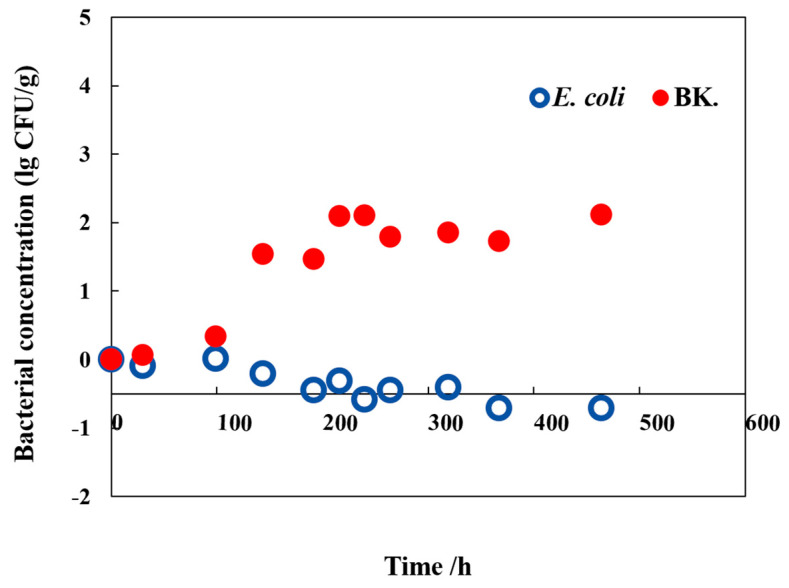
Growth of *E. coli*-R168 (*E. coli*) and background microflora (BK.) in hydroponically grown lettuce at 4 °C.

**Figure 4 foods-13-01359-f004:**
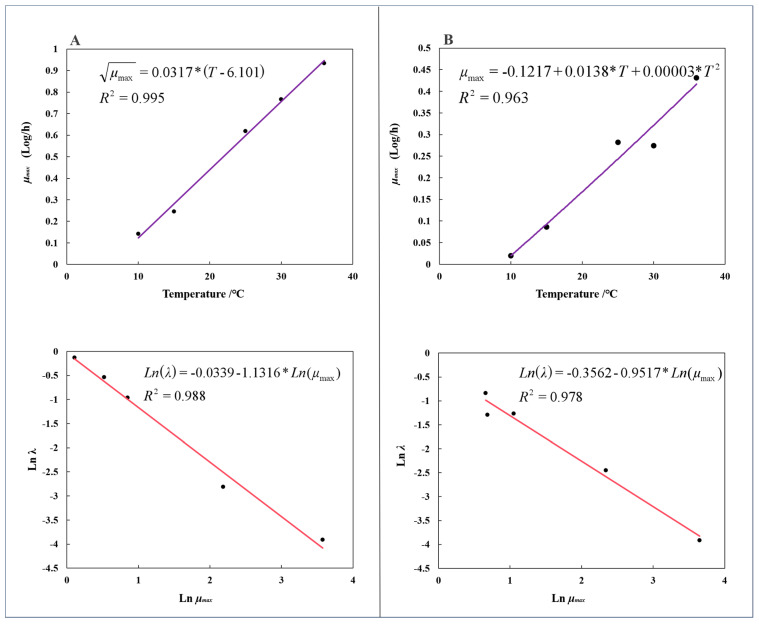
Effect of temperature on the specific growth rates (*μ*_max_) and the lag time of *E. coli*-R168 (**A**) and background microflora (**B**) in hydroponically grown lettuce.

**Figure 5 foods-13-01359-f005:**
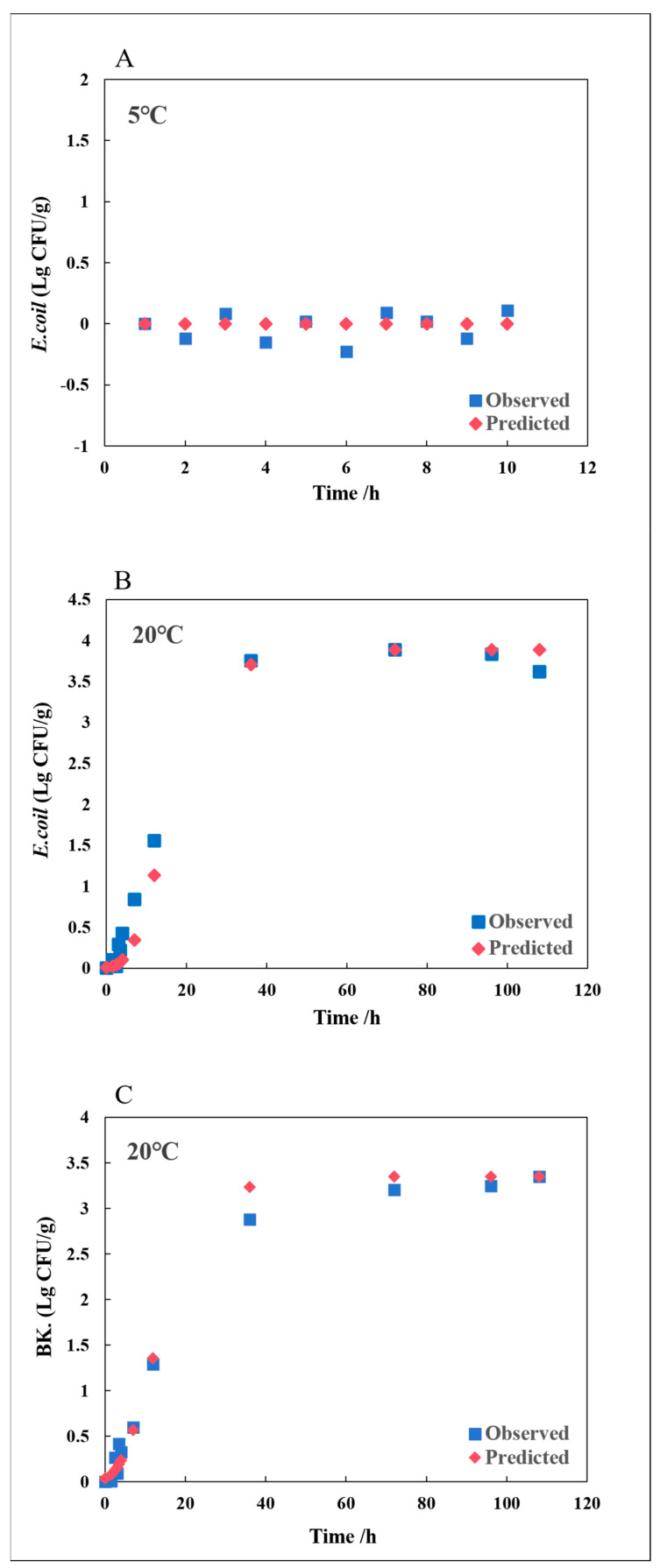
Model validation: growth curves of *E. coli* and background microflora (BK.) in hydroponically grown lettuce at 5 and 20 °C. (**A**) = *E. coli* (from ComBase) at 5 °C; (**B**) = *E. coli* (this study) at 20 °C; (**C**) = background microflora (this study) at 20 °C.

**Table 2 foods-13-01359-t002:** Secondary modeling for growth parameters of *E. coli*-R168 and background microflora in hydroponically grown lettuce.

Bacteria	Secondary Model	Equation	*A_f_*	*B_f_*	RMSE	R^2^
*E. coli*-R168	Ratkowsky square root model	$\sqrt{\mu_{\max}}=0.0317*\left(T-6.101\right)$	1.144	0.989	0.018	0.995
	Polynomial	$\mu_{\max}=0.243-0.266*T+0.0877*T^{2}$	172.629	172.629	60.475	0.963
	Inverse second order	$\mu_{\max}=1.9062+48.019*\left(1/T\right)+292.91*{\left(1/T\right)}^{2}$	1.832	0.737	0.057	0.969
Background microflora	Ratkowsky square root model	$\sqrt{\mu_{\max}}=0.0191*\left(T-0.675\right)$	1.236	1.051	0.002	0.954
	Polynomial	$\mu_{\max}=-0.1217+0.0138*T+0.00003*T^{2}$	1.091	1.000	0.028	0.963
	Inverse second order	$\mu_{\max}=0.785-16.55*\left(1/T\right)+89.145*{\left(1/T\right)}^{2}$	1.117	1.014	0.032	0.952

**Table 3 foods-13-01359-t003:** Validation for the performance of developed models in hydroponically grown lettuce stored at 5 and 20 °C.

Bacteria	Temperature/°C	RMSE	Equation
*E. coli*-R168	5	0.115	$\sqrt{\mu_{\max}}=0.0317*\left(T-6.101\right)$
	20	0.239	$Ln(\lambda )=-0.0339-1.1316*Ln(\mu_{\max})$
Background microflora	20	0.145	$\mu_{\max}=-0.1217+0.0138*T+0.00003*T^{2}$
			$Ln(\lambda )=-0.3562-0.9517*Ln(\mu_{\max})$

## Data Availability

The original contributions presented in the study are included in the article, further inquiries can be directed to the corresponding author.

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
