# Peer review of "Modeling Growth Kinetics of *Escherichia coli* and Background Microflora in Hydroponically Grown Lettuce"

_foods, 2024, doi:10.3390/foods13091359_

Round 1

Reviewer 1 Report

Comments and Suggestions for Authors

The paper describes a straightforward study on developing a model to predict health risk of keeping hydroponic cultivated lettuce at different temperatures. A non-pathogenic Escherichia coli strain was an appropriate choice as representative of a foodborne pathogen seen with lettuce. A model discrimination based on proper statistical analysis was performed for both the growth kinetics and prediction model. The English language is sound, thus does not need improvement.

Here are (mostly) minor comments

1. line 93. Remove ‘l’ from the last word in the sentence ‘representative’.

2. Line 117. Typo: ‘experiments’

3. Lines 136-137. Grammar:  a, b, c are the regression coefficients.

4. Line 139. After µmax, and a and b are regression etc.

5. Lines 149-150. Refer here to Figure 1. This reference is currently missing.

6. Line 162. ‘rangees’ should be ‘ranges’

7.Figure 2A. Top panel should have the same colour code as the other panels.

8. Table 1. Background flora: at 30°C the SLogistic data are missing. Right?

9. Line 195. Typo: ‘s’ missing in ‘superior’

10. Line 205-211. Paragraph. Figure 3 might indicate that the background flora are either inhibited (as the authors observe) or limited in their growth, e.g. due to lacking nutrients or competition with E. coli for nutrients.

11. Line 218. “Didn’t” should be “did not”

12. Line 227. There should be a space between ‘beef’ and ‘and’.

13. Line 241. Typos: Space needed between ‘as’ and ‘native’; ‘there by’ is one word: ‘thereby’

14. References. Names of microorganisms should be in italics.

Reviewer 2 Report

Comments and Suggestions for Authors

I believe that the research topics undertaken by the authors of the publication are very important. Overall, I rate the prepared material very highly.
However, I have a few comments that will help improve the publication.

Sample Preparation and Inoculation

I have doubts whether homogenizing the lettuce before inoculating it with bacterial cultures did not affect the results. The possibilities of bacterial growth in homogenized material are completely different than on lettuce leaves. I ask the authors to respond to this comment so as to indicate the suitability of testing at this low temperature..

Growth studies

It seems that incubation at 4 C was not necessary because it is known that this is too low a temperature for the growth of E. coli.

Results and discussion

No reference in Fig. 1.

Model validation

The authors used data from ComBase for validation. Please provide more information on what specific data were used. The differences are quite significant.

Conclusions

The applications lacked information on the possible storage time. Primary Models should also be used.
